# Carboxypeptidase A4: A Biomarker for Cancer Aggressiveness and Drug Resistance

**DOI:** 10.3390/cancers17152566

**Published:** 2025-08-04

**Authors:** Adeoluwa A. Adeluola, Md. Sameer Hossain, A. R. M. Ruhul Amin

**Affiliations:** 1Department of Pharmaceutical Sciences, School of Pharmacy, Marshall University, Huntington, WV 25755, USA; adeluola.1@buckeyemail.osu.edu (A.A.A.); hossain7@marshall.edu (M.S.H.); 2Division of Pharmaceutics and Pharmacology, College of Pharmacy, The Ohio State University, Columbus, OH 43210, USA

**Keywords:** carboxypeptidase, biomarker, carboxypeptidase A4, drug resistance

## Abstract

Cancer is the second leading cause of death in developed countries and the leading cause of death in the underdeveloped world with about 20 million new diagnoses and 9.7 million deaths worldwide. Identification of biomarkers governing tumor initiation, early diagnosis, aggressiveness, or drug response and resistance is critical to reduce the cancer burden and increase patient survival. Carboxypeptidases, including CPA4, are enzymes that cleave C-terminal peptide bonds in peptides and proteins. Many of these enzymes take part in carcinogenesis and drug response. CPA4 is an exopeptidase that cleaves peptide bonds at the C-terminal domain of peptides and proteins. This enzyme is overexpressed in many cancers and often associated with cancer aggressiveness and resistance to treatment. Targeting CPA4 could serve as a novel strategy to reduce cancer burden and improve drug response.

## 1. Introduction

Peptidases are a class of proteolytic enzymes responsible for the catalytic cleavage of peptide bonds within peptides and proteins. This cleavage is an irreversible, physiological process that occurs when peptides or proteins are processed during digestion, tissue remodeling, and foraging for abnormal proteins and peptides [1]. Peptidases may cleave the internal or the terminal residues of the substrate peptides [2]. Based on these functions, they are classified as endopeptidases (cleave internal peptide bonds) or exopeptidases (cleave terminal peptide bonds). Exopeptidases could either cleave the N-terminal or C-terminal amino acid residue of the substrates, further classifying them into aminopeptidases and carboxypeptidases (CP). Furthermore, some peptidases require a metal ion, e.g., divalent metal ions like zinc, cobalt, manganese, or nickel, to catalyze proteolysis and are called metallopeptidases [2]. Figure 1 illustrates the classification of peptidases.

CPs are a diverse family of enzymes, and their classification is based on a combination of factors that reflect their evolutionary relationships, structural features, and functional properties [3]. There are several classifications for CPs based on substrate specificity, catalytic mechanisms, structure and homology, cellular localization and functions, etc.

Substrate Specificity: This refers to the specific amino acid residues that a CP prefers to cleave from the C-terminus of a protein or a peptide. Based on substrate specificity, CPs are classified into several subclasses: (1) Carboxypeptidase A (CPA): CPAs are zinc containing metallopeptidases that preferentially cleave aromatic or branched-chain amino acids (like phenylalanine, tryptophan, or leucine) at the C-terminus of peptides or proteins. Examples include CPA1, CPA2, and CPA4 [4]. (2) Carboxypeptidase B (CPB): These are also zinc containing metallopeptidases that preferentially cleave basic amino acids such as arginine and lysine residues at the C-terminus of peptides or proteins. (3) Carboxypeptidase E (CPE): CPEs preferentially cleave C-terminal basic amino acid residues, such as lysine and arginine, and are involved in the processing of prohormones and neuropeptides/neurotransmitters [5]. (4) Carboxypeptidase N (CPN): CPNs are zinc containing metallopeptidases present in the blood and cleave basic amino acids from the C-terminus of peptides and proteins involved in inflammation such as anaphylatoxins and kinins. (5) Others: There are other CPs with more specialized substrate specificities, such as those that cleave specific dipeptides or those that act on modified amino acids.

Catalytic Mechanism: Based on the chemical mechanism by which a carboxypeptidase cleaves the peptide bond, they are classified into: (1) Metallocarboxypeptidases: These enzymes utilize a zinc ion in their active site for catalysis. Many carboxypeptidases belong to this category [1]. (2) Serine Carboxypeptidases: These enzymes use a serine residue in their active site for catalysis. They are less common than metallocarboxypeptidases [6]. (3) Cysteine Carboxypeptidases: These enzymes utilize a cysteine residue for catalysis.

Cellular Localization and Function: Based on the cellular location and their role in biological processes, CPs are classified as (1) Secreted Carboxypeptidases: These enzymes are secreted from cells and function extracellularly. Examples include digestive enzymes like CPA1 and CPB. (2) Membrane-bound Carboxypeptidases: These enzymes are anchored to cell membranes and function at the cell surface. (3) Cytosolic Carboxypeptidases: These enzymes are found within the cytoplasm of cells and participate in intracellular processes.

In the current review, we will briefly discuss the history, structure and the role of CPA4 in carcinogenesis and drug resistance.

## 2. CPA4: History and Protein Structure

Carboxypeptidase A4 (CPA4), encoded by the gene *CPA4*, is a zinc containing metallocarboxypeptidase that cleaves amino acids from the C-terminus of peptides and proteins, particularly those with hydrophobic residues, such as Phe, Leu, Ile, Met, Tyr, and Val. The *CPA4* gene was first cloned in prostate cancer cells by mRNA differential display after treating prostate cancer cells with butyrate, a naturally produced short chain fatty acid by bacterial fiber fermentation within the colon and known to regulate cell growth and differentiation of prostate cancer, breast cancer, pancreatic cancer, and hematopoietic cells [7]. Within 6 h of treatment, signs of induction were already evident and continued to 48 h, maximum timepoint tested. The CPA4 full length cDNA contains a short 5′-UTR (7 bp), a large 3′ UTR (1522 bp), and a 1266 bp open reading frame encoding a 421-amino acid protein. The new gene was initially named CPA3 but later renamed CPA4 to maintain the chronological order of discovery of the CPA family members [7,8,9]. The overall structure consists of three main domains: (1) Signal Peptide: This 16-amino acid N-terminus directs the newly synthesized protein to the endoplasmic reticulum for processing and secretion. (2) Pro-domain: The 95 amino acid domain plays a regulatory role, maintaining the enzyme in an inactive state until it is cleaved off. It also assists in the proper folding of the catalytic domain [2]. (3) Catalytic Domain or CPA Domain: This is the core of the enzyme consisting of 310 amino acid residues. It contains the active site where the enzymatic reaction takes place. It has a characteristic α/β hydrolase fold, a common structural motif found in many hydrolytic enzymes [1]. The active site contains a zinc ion (Zn^2+^) that is essential for catalysis. This zinc ion is coordinated by amino acid residues within the catalytic domain, typically histidine and glutamates [1]. Sequence analysis reveals that CPA4 shares significant homology with other members of the carboxypeptidase A/B subfamily, particularly CPA1, CPA2, and CPA3, especially within the catalytic domain [3,4]. It has 37–63% amino acid similarity with zinc CPs from several mammalian species such as rat, bovine, pig, dog, mouse, etc., and 27–43% with several nonmammalian species such Fas black fly, *C. elegans*, Humus earthworm, *Drosophila*, cotton bollworm, *E. coli* and fission yeast [7]. CPA4 is synthesized as a zymogen (inactive precursor) called proCPA4, which undergoes endopeptidase-mediated proteolytic cleavage of the pro-domain to become the active enzyme [2]. The activation process can occur either during synthesis within the cells or after secretion into the extracellular environment [2]. Freshly synthesized proCPA4 is translocated to the lumen of the endoplasmic reticulum (ER), where it is either directed to the lysosomal pathway or secreted to the extracellular environment. CPA4 lacks a transmembrane domain and could not be detected in the extracellular matrix, suggesting that CPA4 is predominantly a secretory protein [10]. The zymogen form has a molecular weight of 50 kDa while the active enzyme is 35 kDa [3].

Huang et al. examined the expression of CPA4 mRNA in 16 normal human tissues by Northern blotting but failed to detect expression in heart, brain, placenta, lung, liver, skeletal muscle, kidney, pancreas, spleen, thymus, prostate, testis, ovary, small intestine, colon, and peripheral blood lymphocytes except prostate cancer cell lines [7]. Subsequent RT-PCR analysis revealed very low expression in some of these tissues, including normal prostate, normal pancreas and pancreatic cancer cell lines [7]. The Human Protein Atlas dataset as of 20/07/2025 reveals that CPA4 is expressed in skin, esophagus, cervix, pancreas, vagina, salivary gland, adipose tissue, adrenal gland, breast, spleen, hypothalamus, placenta, spinal cord, heart muscle, tonsil, thymus, etc., (www.proteinatlas.org). Tissue-based map of huma proteome reveals that CPA4 is expressed in a variety of tissues, with high expression observed in the pancreas [11]. Moderate expression of CPA4 is found in the prostate, salivary gland, small intestine, and stomach. Low levels of CPA4 are detected in the brain, lungs, and other tissues, including the bladder, breast, colon, kidney, liver, ovary, and testis.

Latexin is a 25 kD protein discovered in the brain of rats as a putative noncompetitive endogenous inhibitor of metallocarboxypeptidases, including CPA4 [12,13]. As expected, latexin is highly expressed in tissues with low CPA4 expressions, including the heart, prostate, ovary, pancreas, and colon. Some exogenous inhibitors of carboxypeptidases have also been identified in potatoes [14], leeches [15], Ascaris [16], and marine snail *Nerita versicolor* [17], to name a few.

Catalysis Mechanism: Certain residues within the CP domain are responsible for the peptidase activity of CPA4. Within the catalytic domain, Zn^2+^ binds at the His^69^, Glu^72^, and His^196^ residues, following the numbering convention of CPA1 [18]. Also, the residues for substrate binding and positioning include Arg^71^, Arg^127^, Asn^144^, Arg^145^, and Tyr^248^. For catalysis, Arg^127^ and Glu^270^ are involved [7,8]. Figure 2 illustrates the mechanism of CPA4-me-diated cleavage of terminal amide bond.

CPA4 showed more activity at basic pHs (pH: 8–9) and less activity in acidic environments (pH < 6) [10]. Initially, it was thought that CPA4 had no substrate preferences, but recent reports demonstrate that CPA4 preferentially cleaves peptides with hydrophobic C-terminal residues [10]. Specifically, CPA4 exhibits a preference for cleaving peptides with C-terminal Phe, Leu, Ile, Met, Tyr, and Val [10]. However, other characteristics of the substrate also determine CPA4 specificity. For instance, having an aromatic, aliphatic or basic residue at the P1 position increases the chances of CPA4 cleavage, while acidic residues, Proline and Gly in the P1 position of the substrate will detract from its specificity for CPA4 [10,12].

## 3. CPA4 and Cancer

As mentioned earlier, CPA4 was first cloned in prostate cancer cells. The role of CPA4 in cancer development, treatment response and drug resistance has been explored over many years. The role of CPA4 in various cancers is discussed below.

Prostate Cancer: Several studies have demonstrated a link between CPA4 and prostate cancer. The results by Huang and colleagues demonstrated that treatment of prostate cancer cell lines PC3, DU145, and BH1 with sodium butyrate (NaBu), a histone deacetylase (HDAC) inhibitor, increased the expression of CPA4 mRNA in a p21 ^WAF1/CIP1^ transactivation-dependent manner [7]. HDAC inhibitor trichostatin A also induced CPA4 mRNA expression. This indicated that the observed induction was due to hyperacetylation of the *CPA4* gene. Subsequently, genomic studies revealed that the *CPA4* gene was preferentially expressed by the maternal allele (paternally imprinted) and located at the prostate cancer aggressiveness locus on the 7q32 chromosome and was preferentially expressed in adult benign prostatic hyperplasia, but the expression in normal adult human tissues including prostate, ovary, testis, and pancreas was quite low, suggesting thatCPA4 may be related to prostate cancer aggressiveness [12,19,20].

Ross et al. studied the association of CPA4 polymorphisms with prostate cancer risk [12]. Genotyping of 1012 men–including 506 diagnosed with intermediate-to-high risk prostate cancer, and 506 age-, ethnicity- and hospital-matched control–revealed seven single nucleotide polymorphisms (SNPs). One synonymous SNP was eliminated due to difficulty in primers designing and genotyping. The six SNPs used for analysis are listed in Table 1. Among these SNPs, the non-synonymous coding SNP (rs2171492, Cys303Gly) was associated with an increased risk of prostate cancer in young men. CPA4 substrates like neurotensins, granins, and opioid peptides have been associated with increased cell growth, proliferation, and motility, which may lead to the aggressiveness of prostate cancers [10,21].

Pancreatic Cancer (PC): In PCs, Sun and colleagues demonstrated that the overexpression of CPA4 in PC tissues had a significant correlation with tumor progression [22]. It was evident that serum concentrations of CPA4 were significantly higher in PC patients when compared with healthy individuals. Among 150 PC tissues, 130 of them (86.7%) showed positive staining for CPA4 by IHC. The serum CPA4 levels were also significantly higher in PC patients than for healthy controls and positively correlated with TNM stage, lymph node involvement, and distant metastasis [22]. The study also suggested that a cutoff value of 0.3 ng/mL, CPA4 might serve as a better diagnostic marker of PC than CA1999. The above findings were supported by a subsequent study demonstrating that CPA4 was overexpressed in human PC tissues and was significantly higher compared to the corresponding pancreases (36/65, 55.3% vs. 15/65, 23%, *p* < 0.01) [23]. The expression of CPA4 mRNA was also significantly higher in PC tissues compared with the adjacent pancreas (*n* = 18, *p* = 0.014). Furthermore, CPA4 overexpression in PC tissues was positively related with tumor size (*p* = 0.026), T stage (*p* = 0.011), lymph node metastasis (*p* = 0.026) and worse prognosis (*p* = 0.001). Their mechanistic studies revealed that CPA4 might induce EMT by activating the PI3K/AKT pathway, which promoted motility and invasiveness of PC cells.

Breast Cancer: In breast cancer (BC), studies have shown that MCF-7 cells, a common hormone receptor-positive BC cell line, tend to secrete more CPA4 compared to human mammary epithelial cells (HMEpC) [24]. Interestingly, a unique O-glycosylation pattern of CPA4 was observed in MCF-7 cell media but was absent in HMEpC media, further suggesting a potential role for CPA4 in breast cancer development.

Focusing on triple-negative breast cancer (TNBC) cells, a particularly aggressive subtype of breast cancer, UALCAN dataset analysis revealed that CPA4 mRNA levels were elevated in TNBC, especially in the TP53-mutant subgroup, which were found to be critical for both cell growth and migration, leading to a poorer prognosis for patients in this sub-population [25]. Furthermore, CPA4 overexpression is positively correlated with ALDH1A1, a marker of breast cancer stem cells, and negatively correlated with p53 expression in TNBC. This finding suggests that CPA4 may contribute to the development and maintenance of breast cancer stem cell phenotypes, which are known to drive tumor growth, metastasis, and drug resistance. CPA4 associated aggressive phenotype and poor prognosis in TNBC was further supported by another study conducted by Hanada T et al., [26]. IHC staining showed CPA4 expression was higher in BC tissues (*n* = 221) compared to normal breast tissues. High CPA4 expression was associated with high ALDH1 expression. Furthermore, high-CPA4-expressing TNBC was associated with worse overall and disease-free survival than low-CPA4-expressing TNBC. However, the prognostic significance of CPA4 expression on overall survival could not be validated in a public database cohort. Although high CPA4 expressing patients exhibited poorer prognoses than those with low CPA4 expression, the differences were not significant. This study also confirmed that CPA4 is important for proliferation and migration of TNBC cell lines.

Liu X et al. demonstrated that angiopoietin (ANG1) promotes TNBC cell proliferation by inducing the expression of CPA4 [27]. Using BC samples, this study also confirmed that CPA4 expression was higher in TNBC than non-TNBC and positively correlated with the expression of ANG1. Furthermore, higher CPA4 expression was associated with poor prognosis. However, these findings were contrasted by a study from Bademler and colleagues, who surprisingly found that serum protein and gene expression levels of CPA4 in breast cancer patients were significantly lower than in healthy controls [28]. This discrepancy highlights the complexity of CPA4’s role in breast cancer and underscores the need for further investigation. More studies are required to fully elucidate the impact of CPA4 expression and, importantly, the potential benefits of CPA4 inhibition on breast cancer prognosis and treatment response.

Lung Cancer: Different groups have studied the role and significance of CPA4 in lung tumorigenesis. A study demonstrated gene amplification and an increase in CPA4 levels of lung cancer tissues when compared with normal lung tissue, which correlated with poor prognosis [29]. Their analysis of three Oncomine datasets demonstrated increased CPA4 gene copy number in lung cancer tissues than normal lung tissues in TCGA and Weiss datasets and upregulation of CPA4 mRNA in the Hou lung dataset. Furthermore, 120 out of 165 (72.72%) primary lung lesions exhibited positive IHC staining for CPA4 and no staining of adjacent normal tissues. Moreover, the serum levels of CPA4 were higher in non-small cell lung cancer (NSCLC) patients (*n* = 100) than healthy controls (*n* = 80). They concluded that serum CPA4 levels in combination with known NSCLC biomarker, cytokeratin 19 fragment (CYFRA21) could enable timely diagnosis of NSCLC [29]. Quite recently, using novel high-throughput technology, Pal and colleagues identified CPA4 as one of the genes implicated as a migration control factor in NSCLC cells [30]. This indicates that CPA4 may also play a role in NSCLC metastasis. Another group confirmed that CPA4 promotes lung cancer cell growth and may act via AKT-c-MYC pathway activation through AKT phosphorylation [31]. More so, its depletion significantly suppressed tumor growth in vitro and in vivo. Using miRNA networks, a study reported CPA4 as a target of miR-342-3p, which significantly increased resistance to gefitinib by blocking CPA4 [32]. Interestingly, the upregulation of CPA4 led to apoptosis in gefitinib-resistant lung cancer cells and may restore sensitivity in such cases. However, more in vivo studies are needed to verify these results.

Head and Neck Cancer and Esophageal Cancer: When analyzing the effect of imprinted genes on the prognosis of squamous cell carcinoma of the head and neck (SCCHN), Hsu and colleagues observed that higher expression of CPA4 was related to poor survival (*p* < 0.01) in 73 patients with SCCHN [33]. More importantly, they concluded that loss of imprinting may cause tumorigenesis of SCCHN. In esophageal squamous cell carcinoma (ESCC), high expression of CPA4 was found in 58% (87/150) samples, which were significantly associated with histologic grade, lymph node metastasis, and TNM stage [34]. Furthermore, expressions of ALDH1A1 positively correlated with CPA4 and their co-expression could be used as an independent prognostic factor in ESCC [34].

Gastric and Colon Cancer: Overexpression of CPA4 was observed in about two-thirds of primary gastric cancer tissues (*n* = 100), which correlated positively with Ki-67 and negatively with tumor suppressor p53 [35]. Further analysis showed that the elevated expression of CPA4 in gastric cancers was significantly associated with tumor size, stage, lymph node metastasis, depth of invasion and distant metastasis, suggesting its use as a biomarker for the prognosis of the disease. A study by Lei et al. demonstrated that knocked down of CPA4 by shRNA inhibited colony formation, proliferation, migration and invasion while increased apoptosis of gastric cancer cells [36]. CPA4 expression was also found in about two-thirds (130/190) of colorectal cancer tissues, which significantly correlated with invasion, lymph node metastasis, as well as distal metastasis [37]. Moreover, high CPA4 expression was associated with poor overall survival and is an independent prognostic factor. Similarly, the concentration of CPA4 in serum was significantly increased in CRC patients, and higher serum CPA4 levels were correlated with poor prognosis and liver metastasis in colorectal cancer patients [37]. Future studies could evaluate the inhibition of these pathways to mitigate CPA4 overexpression in colorectal cancer.

Liver Cancer: CPA4 could play a significant role in liver tumorigenesis and cancer stem cell proliferation. 57/100 (57%) liver cancer samples expressed elevated levels of CPA4, and the increased expression of CPA4 significantly correlated with grade, stage and liver cancer stem cell marker CD90 [38]. Both CPA4 and CD90 could serve as independent predictive factors of poor prognosis in hepatic cancer. Moreover, another group confirmed that CPA4 positively influenced the expression of stem cell markers CD133, ALDH1, and CD44 in hepatocellular carcinoma cells. It also induced cell sphere formation in vitro and tumor growth in vivo [39].

Other Cancers: CPA4 also plays crucial role in other cancers, including endometrial cancers, clear cell renal cell carcinoma (ccRCC), bladder cancer and anaplastic thyroid cancer (ATC). Analyzing CPA4 mRNA expressions from TCGA and GEO databases and further validating the results with 116 clinical samples, He et al. reported that CPA4 is significantly upregulated in endometrial cancer, which is correlated with tumor progression and poor prognosis [40]. Furthermore, inhibition of CPA4 expressions in cell lines inhibited tumor growth and spread. Similarly, in ccRCC, CPA4 was overexpressed in TCGA, GEO, and clinical samples (*n* = 24), with higher levels linked to advanced TNM stage, poor histological grade, and worse survival outcomes [41]. Furthermore, CPA4 expression is positively associated with several immune cells (e.g., macrophages, Th2, Treg, B cells) and negatively associated with Th17 cells and neutrophils. Significant differences in immune infiltration scores were observed between high and low CPA4 expression groups across various immune cell types, indicating a strong correlation between CPA4 expression and the tumor’s immune activation state [41]. Using conditioned media (CM) from M2 macrophages, Choi et al. reported that such CM significantly increased the proliferation, migration, and invasion of ATC cells [42]. Further proteomic analysis revealed that CPA4 was only detected in CM of M2 macrophages. Furthermore, knockdown of CPA4 abolished the ability of cells to develop xenograft tumor, suggesting that CPA4 stimulates the progression of thyroid cancer by mediating interactions between M2 macrophages and ATC cells and CPA4 can be a new therapeutic target for the treatment of patients with ATC.

Pan Cancer Expression of CPA4: Wang et al. conducted a differential analysis of CPA4 expression in pan-cancer in TCGA database and found high expression of CPA4 in 14 malignancies, including bladder cancer, ccRCC, breast, SCCHN, lung, etc. [41]. More importantly, tumor tissues have significantly high CPA4 expression compared to normal tissues (*p*  < 0.05). On 23/07/2025, we analyzed the TCGA pan cancer dataset available in the cBioPortal website for *CPA4* alterations (www.cbioportal.org). 248/2583 (9.6%) samples showed alterations in *CPA4*, mostly amplification (169 samples, 6.54%) and mRNA high (64 samples, 2.48%), fewer missense mutation (6 patients, 0.23%), and deep deletion (3 patients, 0.12%). 6 patients (0.23%) have multiple alterations. The data was summarized in Table 2 (those with over 5% alterations).

## 4. CPA4 and Signal Transduction

CPA4 is a signaling molecule that plays a pivotal role in various cellular processes, including cell growth, differentiation, apoptosis and serves as a tumor promoter leading to cancer aggressiveness and drug resistance. The various signaling pathways modulated by CPA4 are summarized below and in Table 3 and Figure 3.

G-Protein coupled receptor (GPCR) Signaling: This is one signaling pathway commonly dysregulated in many cancers. Research shows that genes associated with CPA4 are actively involved in GPCR binding, a critical component of cellular signaling that governs tumorigenesis, including proliferation, invasion, and survival [40]. This association suggests that CPA4 may influence cancer stem cell maintenance and progression through modulation of GPCR pathways.

Estrogen Signaling: CPA4’s involvement in estrogenic signaling is particularly important in hormone-responsive cancers such as endometrial and breast cancers. In endometrial cancer, CPA4 is significantly overexpressed, and gene set enrichment analyses have highlighted the estrogen signaling pathway among those associated with CPA4 expression [40]. This association implies that CPA4 may influence tumor progression through interactions with hormone-dependent cellular processes, including those mediated by estrogen receptors. Similarly, in breast cancer, a study found that high CPA4 expression was significantly associated with aggressive phenotypes in TNBC [26]. Furthermore, patients with TNBC exhibiting high CPA4 expression had significantly poorer overall and disease-free survival than those with low CPA4 expression. These findings suggest that CPA4 may contribute to tumor aggressiveness in TNBC through pathways intersecting with hormone signaling mechanisms. Future mechanistic studies are needed to further validate these associations.

PI3K-AKT-mTOR Signaling: Overexpression of CPA4 in pancreatic cancer cells has been shown to activate the PI3K-AKT-mTOR pathway, promoting EMT, enhancing cell proliferation, and increasing drug resistance. Notably, CPA4 was found to co-immunoprecipitate with AKT, suggesting a direct interaction. Inhibition of PI3K with LY294002 reversed these effects, underscoring the pathway’s role in CPA4-mediated oncogenic activities [23].

AKT-cMYC signaling: Functional studies have shown that CPA4 knockdown leads to decreased phosphorylation of AKT and reduced c-MYC expression, resulting in G1-S cell cycle arrest and increased apoptosis. Conversely, overexpression of CPA4 enhances AKT activation and c-MYC levels, promoting cell proliferation and tumor growth in vivo [31].

STAT3 and ERK signaling: CPA4 is known to promote tumor progression through activation of the STAT3 and ERK signaling pathways in various cancers. For example, in anaplastic thyroid cancer (ATC), CPA4 expression was upregulated in response to M2 macrophage stimulation. This upregulation enhanced proliferation, migration, and invasion of ATC cells. Mechanistically, CPA4 activates the STAT3 and ERK pathways, contributing to tumor progression. Knockdown of CPA4 led to decreased phosphorylation of STAT3 and ERK, resulting in suppressed tumor growth and metastasis [42].

Circular RNA circ-CPA4 in cancer: Circular RNAs function as competing endogenous RNAs (ceRNA), interact and sequester mRNAs, thereby modifying the expression of miRNA target genes [43]. CPA4 is a downstream target of let-7, which is a known tumor suppressor [44]. However, in gliomas, their interaction is competitively inhibited by circular CPA4 RNA (circCPA4), which acts as a competitive endogenous RNA (ceRNA). This ceRNA mechanism leads to the upregulation of CPA4 expression and ultimately poor survival outcomes in gliomas [45]. By contrast, circCPA4 also functioned as a ceRNA that inhibited let-7 miRNA expression in NSCLC [46]. This led to increased expression of PD-L1, which is another target of let-7. The increased levels of PD-L1 intracellularly and extracellularly through exosomes, induced stemness characteristics and cisplatin resistance in NSCLC cells by inactivating CD8^+^. Furthermore, Li et al. reported that circRNA-CPA4 is overexpressed in four NSCLC cell lines compared to immortalized bronchial epithelial cells, and that targeting circRNA-CPA4 inhibits proliferation and induces apoptosis of NSCLC cells, as well as in vivo xenograft growth [47]. miR-1183 was identified as the target mi-RNA which regulates the expression of PDPK1/PDK1.

## 5. Conclusions

There is significant evidence from the literature that CPA4 promotes tumor progression by regulating cell proliferation and survival, EMT, invasion and metastasis. It holds strong potential as a diagnostic biomarker and is an actionable target for research on cancer therapeutics. Despite its promise, however, there are still significant gaps in science that require further investigation. Most importantly, the rigor for many findings is weak, supported by one or a few articles. Huang and colleagues reported that CPA4 expression after NaBu treatment was dependent on p21 transactivation, suggesting epigenetic regulation of CPA4 [7]. However, subsequent studies have reported the overexpression of CPA4 in different types of cancer but have done little to explain how it is consistently elevated in these tumors. The hypoxic changes in the tumor microenvironment may be responsible for the frequent overexpression of CPA4 commonly seen in different cancers. A study demonstrated that the overexpression of CPA4 in hypoxic human adipose-derived stem cells was HIF-1α dependent [48]. Some studies have proposed the activation of the PI3K pathway as a possible mechanism for CPA4 activity in cancer cells [23,31], while others propose activation of STAT and ERK pathways [42]. More studies are required to validate these preliminary results and explore the possibility of context-dependent mechanisms. A specific CPA4 SNP was found to be associated with an increased risk of aggressive prostate cancer [11]. Whether these CPA4 genetic variants are sex dimorphic or if they are implicated in other malignancies should be a topic for future studies.

While many studies have reported the association between CPA4 overexpression and poor prognosis, few have explored ways to inhibit it [49]. If truly CPA4 plays a role in tumor aggressiveness and drug resistance, future studies must explore new ways to inhibit CPA4 and assess the effect of inhibition on tumor prognosis. In summary, the challenges include the following: (1) the exact substrates and molecular pathways regulated by CPA4 in cancer are not fully understood; (2) selective targeting of CPA4 without affecting other related carboxypeptidases remains a challenge; and (3) validation of potential role of CPA4 in cancer with multiple studies to ensure rigor. In addition, the role of CPA4 in drug response and drug resistance has not been explored well. There is a paucity of concrete direct mechanistic evidence linking CPA4 to chemoresistance. Although it has been two decades since CPA4’s discovery and its association with cancer development and progression, it is still a very nascent research topic. Future research should aim to (1) unravel CPA’s precise molecular function in cancer and develop targeting strategies for its modulation; (2) in vitro drug response studies following CPA4 knockdown and overexpression; (3) in vivo studies assessing therapeutic response in models with CPA4 modulations (transgenic and knockout mice, xenografts with cells overexpressing or knockout CPA4); and (4) explore combinatorial strategies through CPA4 targeting and other chemotherapy drugs. In ccRCC, CPA4 was found to influence immune cell infiltration, suggesting its potential role in regulating tumor immune microenvironment (TIME), which might impact immunotherapy response. CPA4’s role in TIME and immunotherapy response is another potential area to explore. Studies also demonstrate the potential role of serum CPA4 in predicting metastasis. Therefore, emphasis is needed to explore its application in liquid biopsy as a biomarker for predicting drug response or metastasis. In conclusion, CPA4 represents a compelling link between proteolytic regulation and cancer pathogenesis. As research continues, CPA4 could emerge as a novel tool in the diagnosis and treatment of various malignancies, potentially contributing to more personalized, safe and effective cancer care.

## Figures and Tables

**Figure 1 cancers-17-02566-f001:**
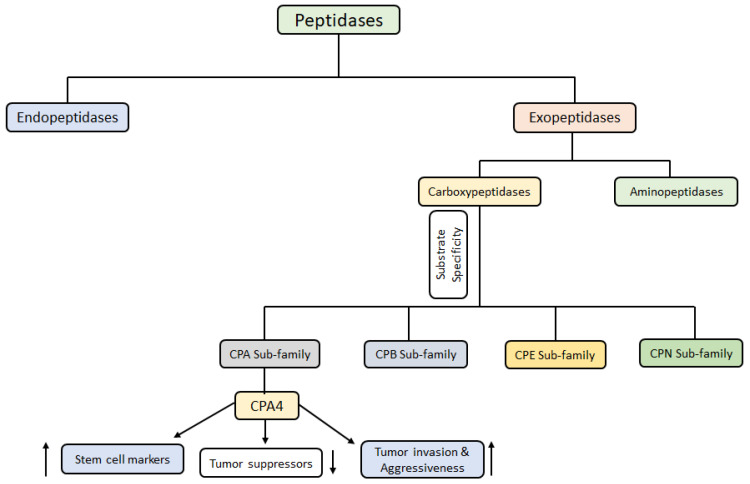
Classification of peptidases.

**Figure 2 cancers-17-02566-f002:**
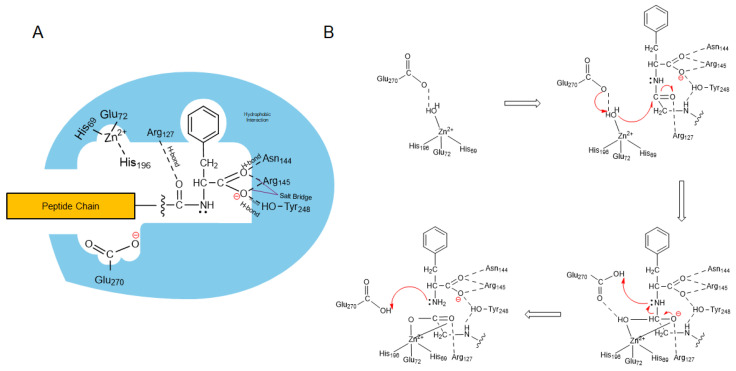
Mechanism of catalysis by CPA4. (**A**) Positioning of substrate within the catalytic pocket of CPA4. (**B**) Catalysis reactions with intermediates. Arg^145^ forms ionic interaction with the carboxylic acid group of the substrate. Arg^127^ stabilizes the negatively charged intermediate during the hydrolysis reaction by forming hydrogen bonds with the oxygen atoms of the carbonyl group of the substrate. Tyr^248^ and Asn^144^ contribute to substrate binding through hydrogen bonding and ion-dipole interactions with the carboxylate group. The complex is further stabilized through hydrophobic interaction with the hydrophobic residue of the C-terminal amino acid. The reaction is initiated with deprotonation and activation of the water molecule by Glu^270^ (electron donor as a base), followed by nucleophilic attack of the carbonyl carbon by activated water (hydroxyl ion). Finally, Glu^270^ acts as an acid, donating a proton to the leaving nitrogen group, facilitating the cleavage of the peptide bond. Red arrows indicate electron transfer and circles highlight negative charge.

**Figure 3 cancers-17-02566-f003:**
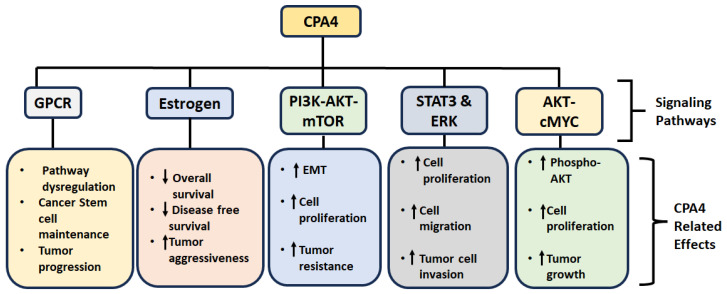
Signaling pathways and associated cellular effects of CPA4 overexpression. Up-facing arrow indicates increase and down-facing indicates decrease.

**Table 1 cancers-17-02566-t001:** List of CPA4 SNPs.

SNPs	Location	Allele Change
rs901799	5′	C > A
rs3807344	intron	A > G
rs1569132	intron	A > G
rs1038628	intron	G > T
rs2171492	exon	A > G
rs1488009	intron	G > T

**Table 2 cancers-17-02566-t002:** Alterations of *CPA4* in pan cancer TCGA dataset.

Cancer Type	Amplification	mRNA High	Mutation	Deep Deletion
Cervical SCC	0 (0%)	10/18 (55.56%)	0/18 (0%)	0/18 (0%)
SCCHN	1/43 (2.33%	21/43 (48.84%)	0/43 (0%)	0/43 (0%)
Ovarian	0/23 (0%)	11/23 (47.83%)	0/23 (0%)	0/23 (0%)
Renal	12/32 (37.5%)	0/32 (0%)	0/32 (0%)	0/32 (0%)
Pancreatic	23/81 (28.4%)	0/81 (0%)	0/81 (0%)	0/81 (0%)
Melanoma	22/107 (20.56%)	0/107 (0%)	1/107 (0.93%)	0/107 (0%)
Leiomyosarcoma	2/15 (13.33%)	1/15 (6.67%)	0/15 (0%)	0/15 (0%)
Bladder SCC	0/23 (0%)	3/23 (13.04%)	0/23 (0%)	0/23 (0%)
Glioblastoma	6/39 (15.38%)	0/39 (0%)	0/39 (0%)	0/39 (0%)
Lung SCC	4/47 (8.51%)	2/47 (4.26%)	1/47 (2.13%)	0/47 (0%)
Breast	4/34 (11.76%)	0/34 (0%)	1/34 (2.94%)	0/34 (0%)
Serous Ovarian	11/100 (11%)	1/100 (1%)	0/100 (0%)	0/100 (0%)
Stomach Adenocarcinoma	2/17 (11.76%)	0/17 (0%)	0/17 (0%)	0/17 (0%)
Uterine	1/20 (5%)	1/20 (5%)	0/20 (0%)	0/20 (0%)
Chromophobe Renal	4/43 (9.3%)	0/43 (0%)	0/43 (0%)	0/43 (0%)
Esophageal	8/97 (8.25%)	0/97 (0%)	0/97 (0%)	1/97 (1.03%)
Papillary Stomach Adenocarcinoma	2/22 (9.09%)	0/22 (0%)	0/22 (0%)	0/22 (0%)
Colorectal	4/44 (9.09%)	0/44 (0%)	0/44 (0%)	0/44 (0%)
Medulloblastoma	7/93 (7.53%)	0/93 (0%)	1/93 (1.93%)	0/93 (0%)
Osteosarcoma	3/35 (8.57%)	0/35 (0%)	0/35 (0%)	0/35 (0%)
Uterine	2/24 (8.33%)	0/24 (0%)	0/24 (0%)	0/24 (0%)
Cholangiocarcinoma	2/26 (7.69%)	0/26 (0%)	0/26 (0%)	0/26 (0%)
Anaplastic Medulloblastoma	2/27 (7.41%)	0/27 (0%)	0/27 (0%)	0/27 (0%)
ccRCC	4/111 (3.6%)	4/111 (3.6%)	0/111 (0%)	0/111 (0%)
Burkit Lymphoma	1/17 (5.88%)	0/17 (0%)	0/17 (0%)	0/17 (0%)
Oligodendroglioma	1/18 (5.56%)	0/18 (0%)	0/18 (0%)	0/18 (0%)
Liposarcoma	1/19 (5.26%)	0/19 (0%)	0/19 (0%)	0/19 (0%)
HCC	13/306 (4.25%)	1/306 (0.33%)	1/306 (0.33%)	1/306 (0.33%)

**Table 3 cancers-17-02566-t003:** Summary of CPA4’s involvement in different cell signaling pathways.

Signaling Pathway	CPA4 Role	Key Findings (Reference)
GPCR Signaling	Associated with GPCR binding genes	CPA4 expression correlates with GPCR-related functions in cancer stem cell biology [40]
Estrogen Signaling	Correlated with hormone-responsive tumors	Overexpressed in endometrial cancer; implicated in ER-positive cancer processes [26,40]
PI3K-AKT-mTOR	Activates pathway components	Enhances EMT, drug resistance, and proliferation in pancreatic and cardiac tissues [23]
AKT–cMYC	Promotes AKT phosphorylation and c-MYC expression	Drives proliferation and inhibits apoptosis in NSCLC [31]
STAT3 and ERK	Activates both STAT3 and ERK	Enhances migration, invasion, and proliferation in thyroid cancer [42]

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
