# Peer review of "Carboxypeptidase A4: A Biomarker for Cancer Aggressiveness and Drug Resistance"

_cancers, 2025, doi:10.3390/cancers17152566_

Round 1

Reviewer 1 Report

Comments and Suggestions for Authors

The review titled “Carboxypeptidase A4: A Biomarker for Cancer Aggressiveness and Drug Resistance” by Adeluola et al. offers a valuable, well-structured overview of CPA4—examining its classification, structural characteristics, expression in normal tissues, and its roles across various cancer types, including the cellular processes and signaling pathways it modulates. This is the first comprehensive review exploring CPA4’s link to cancer. Overall, the manuscript is thoughtfully composed: it draws from the majority of relevant studies, includes clear and informative tables and figures, and persuasively demonstrates that CPA4 is frequently overexpressed in numerous cancers, correlating with tumor aggressiveness and potential biomarker utility. However, the evidence supporting CPA4’s role in drug resistance remains limited in rigor and depth.

To strengthen the Future Directions section of the review, I recommend the authors explicitly address this weakness by:

  1. Highlighting the lack of direct mechanistic evidence linking CPA4 to chemoresistance.
  2. Proposing future studies involving:
    • In vitro drug‑response assays following CPA4 knockdown/overexpression.
    • In vivo models assessing therapeutic response once CPA4 is modulated.
    • Investigation into combination therapies targeting CPA4 to overcome resistance.

In addition, need to add further depth:

  • Immune microenvironment interactions: Emerging research—such as in clear cell RCC—suggests CPA4 may influence immune cell infiltration, meriting discussion around its potential impact on immunotherapy response.
  • Liquid‑biopsy applications: Serum CPA4 measurements, particularly when combined with established markers (e.g., CYFRA21‑1 in NSCLC, AUC ≈ 0.83; CRC liver metastasis prediction, AUC ≈ 0.96), represent a promising translational direction that warrants emphasis.

Manuscript offers a significant, timely contribution. By bolstering the discussion around drug resistance, immune context, and non-invasive diagnostics, it will provide readers with both clarity and a roadmap for future research.

Reviewer 2 Report

Comments and Suggestions for Authors

This manuscript presents a comprehensive review of the role of Carboxypeptidase A4 (CPA4) in cancer biology and its involvement in tumor aggressiveness and drug resistance. The authors have compiled a large range of studies across multiple cancer types, highlighting CPA4’s potential as a biomarker and therapeutic target. The review is timely and relevant, given the growing interest in proteolytic enzymes in cancer progression.

The manuscript provides a thorough and well-structured examination of CPA4’s involvement in various cancers, including prostate, pancreatic, breast, head and neck, esophageal, Colon, lung, gastric, and liver malignancies. The authors highlighting CPA4’s influence on key molecular pathways such as PI3K-AKT-mTOR, STAT3-ERK, and GPCR signaling. These insights are clearly presented making the review easy to read and understand.

Minor Issues

Reference Duplication: please review references 7 and 9

Reviewer 3 Report

Comments and Suggestions for Authors

The manuscript “Carboxypeptidase A4: A Biomarker for Cancer Aggressiveness and Drug Resistance” is a review article about Carboxypeptidase A4, an exopeptidase that cleaves peptide bonds at the C-terminal domain within peptides and proteins, and its role in cancer phenotype and drug resistance.

The manuscript may be interesting for the readers but contains several crucial flaws that prevent its publication. I was unsure whether to reject the manuscript or request major revisions; after careful consideration, I have opted for the latter. Authors are required to address the following concerns:

  1. The manuscript requires English grammar revision (mandatory).
  2. The abstract needs to be more thoroughly revised to more clearly emphasize the main theme of this review.
  3. Authors should underline the novelty of the present review compared to existing literature.
  4. I have serious doubts that the topic justifies a review. In fact, the number of experimental articles that focused on this subject do not justify a literature review.
  5. The authors should report the limitations of the cited studied and provide a critical evaluation of the reported findings.
  6. A figure describing the Carboxypeptidase A4 mechanism of action is mandatory.
  7. The paragraph 2 about protein structure should be expanded providing all details available about enzyme structure, localization, tissue expression kinetics, etc..
  8. The manuscript would benefit from RNA-seq database analysis and graph to evidence highest and lowest expression profiles across tumor types.
  9. Please report known important polymorphisms of the gene associated with enzyme activity.
  10. Please evidence mechanisms impacted by Carboxypeptidase A4 activity by a specific paragraph and illustration.
  11. The conclusion section should better explain challenges and future perspectives.

Round 2

Reviewer 3 Report

Comments and Suggestions for Authors

The manuscript has been improved and can be accepted for publication